# Triterpenoids from Kochiae Fructus: Glucose Uptake in 3T3-L1 Adipocytes and *α-*Glucosidase Inhibition, In Silico Molecular Docking

**DOI:** 10.3390/ijms24032454

**Published:** 2023-01-26

**Authors:** Xue-Lin Chen, Kun Zhang, Xia Zhao, Han-Lei Wang, Mei Han, Ru Li, Zhen-Nan Zhang, Yu-Mei Zhang

**Affiliations:** 1Key Laboratory of Tropical Plant Resource and Sustainable Use, Xishuangbanna Tropical Botanical Garden, Chinese Academy of Sciences, Kunming 650223, China; 2School of Life Science, University of Chinese Academy of Sciences, Beijing 100049, China

**Keywords:** Kochiae Fructus, *Kochia scoparia*, triterpenoids, *α*-glucosidase, glucose uptake, molecular docking

## Abstract

In this study, three new triterpenes (**1**–**3**) and fourteen known triterpenoids (**4**–**17**) were isolated from the ethanol extract of Kochiae Fructus, and their structures were elucidated by analyzing UV, IR, HR-ESI-MS, 1D, and 2D NMR spectroscopic data. Among them, compounds **6**, **8**, and **11**−**17** were isolated for the first time from this plant. The screening results of the glucose uptake experiment indicated that compound **13** had a potent effect on glucose uptake in 3T3-L1 adipocytes at 20 μM. Meanwhile, compounds **3**, **9** and **13** exhibited significant inhibitory activities against *α*-glucosidase, with IC_50_ values of 23.50 ± 3.37, 4.29 ± 0.52, and 16.99 ± 2.70 µM, respectively, and their *α*-glucosidase inhibitory activities were reported for the first time. According to the enzyme kinetics using Lineweaver–Burk and Dixon plots, we found that compounds **3**, **9** and **13** were *α*-glucosidase mixed-type inhibitors with *K_i_* values of 56.86 ± 1.23, 48.88 ± 0.07 and 13.63 ± 0.42 μM, respectively. In silico molecular docking analysis showed that compounds **3** and **13** possessed superior binding capacities with *α*-glucosidase (3A4A AutoDock score: −4.99 and −4.63 kcal/mol). Whereas compound **9** showed +2.74 kcal/mol, which indicated compound **9** exerted the effect of inhibiting *α*-glucosidase activity by preferentially binding to the enzyme−substrate complex. As a result, compounds **3**, **9** and **13** could have therapeutic potentials for type 2 diabetes mellitus, due to their potent hypoglycemic activities.

## 1. Introduction

Diabetes mellitus is a chronic metabolic illness that has become a global public health problem [1]. According to the International Diabetes Federation (IDF), approximately 536.6 million adults between the ages of 20 and 79 years had diabetes mellitus in 2021 and this number is expected to rise to 783.2 million by 2045 [2]. Type 2 diabetes mellitus (T2DM) accounts for over 90% of all patients with diabetes and is characterized mainly by insulin resistance, reduction of insulin secretion, and hyperglycemia [3]. It is well established that decreased peripheral glucose uptake, combined with augmented endogenous glucose production, are characteristic features of insulin resistance [4]. Glucosidase enzymes catalyze hydrolysis of starch into simple sugars. In humans, these enzymes aid digestion of dietary carbohydrates and starches to produce glucose for intestinal absorption, which in turn, leads to an increase in blood glucose levels [5]. From these perspectives, enhancing the glucose uptake of organs or tissues [6], as well as inhibiting the activity of *α*-glucosidase [7], are major strategies for T2DM patients to maintain proper blood glucose levels.

Adipocytes play a vital role in glucose metabolism. 3T3-L1 preadipocytes, after differentiation to adipocytes, serve as excellent in vitro models and are useful tools in understanding the glucose metabolism [8]. Kim SH previously proved that berberine-activated, GLUT1-mediated glucose uptake in 3T3-L1 adipocytes [9]. Recent research by Shyni GL found that chebulagic acid from *Terminalia chebula* enhanced insulin-mediated glucose uptake in 3T3-L1 adipocytes via the PPARγ signaling pathway [10]. This research provided proof that the 3T3-L1 adipocyte glucose uptake model was an effective and feasible way for T2DM research. Furthermore, early studies have shown that the inhibition of *α*-glucosidase activity could retard the absorption of glucose and decrease the postprandial blood glucose levels [11,12]. Therefore, *α*-glucosidase has been taken as a key target for treating diabetes, and the inhibitors of *α*-glucosidase can be developed into effective therapeutic drugs to treat T2DM [13].

Kochiae Fructus (KF) is the dried fruits of *Kochia scoparia*, which is an important Chinese herbal medicine. It contains a variety of bioactive components such as triterpenoids [14,15], flavonoids [16,17], carbohydrates [18], amino acids [18], organic acids [18], and essential oils [19]. KF was listed as ‘top grade’ medicinal material by the *Shennong’s Classic of Materia Medica* and has been used in traditional Chinese medicine to treat rubella, eczema, cutaneous pruritus, and difficulty and pain in micturition [20,21]. Modern pharmacological research showed that KF possessed a broad range of pharmacological activities, including anti-inflammatory [22], hypoglycemic [23], antioxidant [24], anticancer [25], antifungal [26], anti-pruritogenic [27], antinociceptive [28], antiallergic [29], and hepatoprotective [30] activity.

According to literature, the antidiabetic activity of the n-butanol fraction of KF had been investigated, however, there are few reports on the hypoglycemic mechanism of corresponding chemical constituents of KF. Therefore, in this study, chemical constituents of KF were isolated and identified. Furthermore, the glucose uptake assay and *α*-glucosidase inhibition experiment were used to explore its hypoglycemic effect and the possible hypoglycemic mechanism. In addition, in silico molecular docking study was used to determine the inhibitory activities of candidate compounds against *α*-glucosidase. Herein are reported the isolation and structure determination of three new triterpenes (**1**−**3**) along with fourteen known triterpenoids (**4**−**17**) from the ethanol extract of KF. Among them, compounds **6**, **8**, and **11**−**17** were isolated for the first time from this plant. In addition, the glucose uptake effect of compound **13** and the inhibitory activities of compounds **3**, **9**, and **13** against *α*-glucosidase were reported for the first time.

## 2. Results and Discussion

### 2.1. Chemistry

Compound **1** was obtained as a white amorphous powder. The molecular formula was established to be C_41_H_62_O_15_ (eleven degrees of unsaturation) by HR-ESI-MS at *m*/*z* 396.1971 [M-2H]^2−^ (calcd 396.1972), which was confirmed by ^13^C and DEPT NMR spectra. The ^13^C NMR spectrum (Table 1) revealed signals due to six methyl carbons at *δ*_C_ 33.7 (C-29), 26.4 (C-27), 24.1 (C-30), 17.7 (C-26), 16.4 (C-25) and 12.2 (C-24); one oxygen-bearing methine carbon at *δ*_C_ 86.6; a pair of olefinic carbons at *δ*_C_ 123.6 (C-12) and 145.3 (C-13); and two carboxylic acid carbons at *δ*_C_ 181.9 (C-28) and 181.5 (C-23). These data, coupled with corresponding information from the ^1^H NMR spectrum [six methyl groups as singlets at *δ*_H_ 0.80 (H-26), 0.90 (H-29), 0.93 (H-30), 0.96 (H-25), 1.14 (H-24) and 1.16 (H-27); hydroxymethine proton at *δ*_H_ 4.05 (dd, *J* = 11.5, 4.0 Hz, H-3); a proton attributed to H-18 at *δ*_H_ 2.84 (dd, *J* = 13.5, 4.0 Hz); and a triplet vinyl proton at *δ*_H_ 5.23 (1H, brs, H-12)] (Table 2) and 2D NMR spectra, confirmed that compound **1** had an oleanolic acid skeleton. In the HMBC spectrum, correlation signals from *δ*_H_ 4.05 (H-3) to *δ*_C_ 26.6 (C-2), 54.0 (C-4), 181.5 (C-23), and 12.2 (C-24) were exhibited, which confirmed the location of the hydroxyl group at C-3. In the ROESY contour map, correlations between H-3 and H-5 were detected, which confirmed that H-3 was *α*-oriented. The aforementioned NMR data of compound **1** were closely similar to those of momordin Ic [31], except for the signals from ring A, a carboxyl group at C-23 position. In the HMBC spectrum, correlation signals from *δ*_H_ 1.14 (H-24) to *δ*_C_ 86.6 (C-3) and 181.5 (C-23) were exhibited, which confirmed the location of the carboxyl group at C-23 (Figure 1). Furthermore, the hypothesis was supported by the correlations between H-25 (*δ*_H_ 0.96), H-24 (*δ*_H_ 1.14), and H-26 (*δ*_H_ 0.80) in the ROESY contour map (Figure 1). From the above data, the structure of compound **1** was deduced to be 3(*β*)-3-*O*-*β*-D-xylopyranosly(1→3) -*β*-D-glucopyranosiduronic acid-12-en-23,28-dioic acid.

Compound **2** was obtained as a white amorphous powder. Its molecular formula was determined to be C_42_H_64_O_15_, on the basis of a negative HR-ESI-MS profile (*m*/*z*: 403.2053 [M-2H]^2−^, calcd 403.2050), implying eleven degrees of unsaturation. The ^13^C NMR (Table 1) and HSQC spectra of compound **2** showed 42 carbon signals comprising ten quaternary carbon atoms [including two carboxylic acid carbon at *δ*_C_ 182.0 (C-28) and 181.7 (C-23)]; fourteen methine carbons [including an olefinic carbon at *δ*_C_ 123.6 (C-12)]; one oxymethine carbon at *δ*_C_ 86.6 (C-3); and two anomeric carbon signals at *δ*_C_ 105.5 (C-glcA-1) and 105.9 (C-xyl-1)]; eleven methylenes; six methyls; and a methoxy carbon (*δ*_C_ 53.0). The ^1^H NMR spectrum of compound **2** (Table 1) exhibited six methyl groups as singlets at *δ*_H_ 0.81 (H-26), 0.91 (H-29), 0.94 (H-30), 0.96 (H-25), 1.14 (H-24) and 1.17 (H-27); hydroxymethine proton at *δ*_H_ 4.05 (dd, *J* = 11.5, 4.5 Hz, H-3); a proton attributed to H-18 at *δ*_H_ 2.85 (dd, *J* = 14.0, 4.0 Hz); a triplet vinyl proton at *δ*_H_ 5.24 (1H, brs, H-12); and a methoxy group at *δ*_H_ 3.77 (3H, s). Comparison of ^1^H and ^13^C NMR values and the analysis of the HMBC correlations showed that compound **2** was entirely consistent with compound **1,** except for one methoxy carbon (*δ*_C_ 53.0, *δ*_H_ 3.77) that occurred in compound **2**. In the HMBC spectrum, correlation signals from *δ*_H_ 3.77 to *δ*_C_ 171.2 (C-glc-6) were exhibited, which confirmed the location of the methoxy group at C-6′ of the glucuronopyranose unit (Figure 2). According to the above data, the structure of compound **2** was elucidated as 3(*β*)-3-*O*-*β*-D-xylopyranosly(1→3) -*β*-D-glucopyranosiduronic acid 6-methyl ester-12-en-23,28-dioic acid.

Compound **3** was obtained as a white powder with a molecular formula of C_36_H_58_O_9_ (eight degrees of unsaturation), as determined by the pseudo-molecular ion peak at *m*/*z*: 657.3973 [M+Na]^+^ (calcd 657.3973) shown in the HR-ESI-MS spectra. The ^1^H NMR spectrum of compound **3** (Table 1) revealed seven angular methyl signals at *δ*_H_ 0.80 (H-26), 0.80 (H-24), 0.91 (H-29), 0.94 (H-30), 1.01 (H-25), 1.01 (H-23) and 1.16 (H-27); one olefinic proton signal at *δ*_H_ 5.27 (1H, brs, H-12); two oxygen-substituted proton signals [*δ*_H_ 3.62 (1H, m, H-2) and 2.90 (d, *J* = 9.5 Hz, H-3)]; and a proton attributed to H-18 at *δ*_H_ 2.86 (dd, *J* = 14.0, 4.0 Hz). The ^13^C NMR (Table 1) and HSQC spectra (Figure 3) of compound **3** showed 36 carbon signals including a carboxylic acid carbon [*δ*_C_ 178.2 (C-28)]; a pair of olefinic carbons [*δ*_C_ 145.2 (C-13) and 123.8 (C-12)]; two oxygenated methine carbons [*δ*_C_ 84.6 (C-3) and 69.6 (C-2)]; one anomeric carbon [*δ*_C_ 95.9 (C-glc-1)]; and seven methyl carbons [*δ*_C_ 33.6 (C-29), 29.4 (C-23), 26.5 (C-27), 24.1 (C-30), 17.9 (C-24), 17.6 (C-26) and 17.3 (C-25)], which were characteristic of the 12-ene triterpene containing two hydroxy groups. The observed HMBC correlations from C-2 (*δ*_C_ 69.6) to H-1 (*δ*_H_ 1.92 and 0.88), H-3 (*δ*_H_ 2.90), and H-23 (*δ*_H_ 1.01), and from H-3 (*δ*_H_ 2.90) to C-1 (*δ*_C_ 48.3), C-2 (*δ*_C_ 69.6), C-4 (*δ*_C_ 40.7), C-23 (*δ*_C_ 29.4), and C-24 (*δ*_C_ 17.9), combined with the ^1^H-^1^H COSY correlations of H-1/H-2/H-3, established the existence of the two hydroxy groups at C-2 and C-3, respectively. The aforementioned NMR data of compound **3** were closely similar to those of oleanolic acid 28-*O*-*β*-D-glucopyranosyl ester [32], except for the configuration of the hydroxy group at C-2. In the ROESY spectrum of compound **3** (Figure 3), cross-peaks were observed between H-2/H-3, H-23; H-3/H-2, H-5, H-23, implied that the configurations of hydroxy groups at C-2 and C-3 were both defined as *β*-orientation. Therefore, the structure of compound **3** was assigned as 2*β*,3*β*- Dihydroxyolean-12-en-28-oic acid 28-*O*-*β*-D-glucopyranoside.

The other fourteen known compounds (Figure 4) were characterized as momordin Ic (**4**) [31], momordin Ic 6’-methyl ester (**5**) [33], momordin Ic ethyl ester (**6**) [34], momordin IIc (**7**) [35], momordin IIc 6’-methyl ester (**8**) [14], 2’-O-β-D-glucopyranosyl momordin IIc (**9**) [33], oleanolic acid (**10**) [36], gypsogenic acid (**11**) [37], oleanolic acid 28-*O*-*β*-D-glucopyranosyl ester (**12**) [32], 3*β*,22*β*-butyryloxy-olean-12-en-28-oic acid (**13**) [38], (3*β*)-3-(*β*-D-glucopyranosyloxy)olean-18-en-28-oic acid (**14**) [39], (3*β*)-3-(*β*-D-glucopyranosyloxy)olean-13(18)-en-28-oic acid (**15**) [40], (3*β*)-3-(*β*-D-glucuronopyranosyloxy)olean-12-en-23,28-dioic acid (**16**) [41], (3*β*)-3-(*β*-D-glucuronopyranosyloxy)olean-11,13-dien-28-oic acid (**17**) [42].

### 2.2. Glucose Uptake and Cell Viability

Using 3T3-L1 adipose model cells, the insulin-induced glucose uptake-enhancing assay was performed to determine glucose consumption after treatment with the compounds. The activity represents the compounds’ potential to reduce insulin resistance in body tissues, such as adipocytes, resulting in hypoglycemic effects [43]. To examine the effects of the isolated compounds from KF with glucose uptake in 3T3-L1 adipocytes, 14 compounds (purity ≥ 95%; HPLC) were screened for their abilities to enhance glucose uptake upon the induction of insulin against fully differentiated 3T3-L1 cells. As shown in Figure 5a, the insulin group could significantly promote the glucose uptake rates of 3T3-L1 adipocytes compared to the control group with a significant difference (*p* < 0.001). Simultaneously, compound **13** had a strong effect on glucose uptake in 3T3-L1 adipocytes at 20 μM (*p* < 0.001). Additionally, the results of cell viability showed that compared to the control group, the isolated compounds had no cytotoxicity (Figure 5b). In conclusion, compound **13** had a significant capability to promote glucose uptake in 3T3-L1 adipocytes, and without inhibitory effects on cell viability.

### 2.3. α-Glucosidase Inhibition Activity

*α*-Glucosidase inhibitors exert hypoglycemic effects by slowing the digestion of carbohydrates and delaying glucose absorption [44]. Identification of potential *α*-glucosidase inhibitors were done by in vitro screening of 14 pentacyclic triterpenes (purity ≥ 95%; HPLC) using an *α*-glucosidase inhibition experiment. The results are shown in Table 3. Compounds **1**–**9**, **11**–**14**, and **17** exhibited varying degrees of *α*-glucosidase inhibitory activity, with inhibitory rates between 13.71 ± 0.54 and 74.41 ± 1.02%. Further tests of *α*-glucosidase inhibitory activities on compounds **3**, **9**, and **13** were carried out, and the results are shown in Figure 6. Compounds **3**, **9**, and **13** showed the most potent *α*-glucosidase inhibition activities (*p* < 0.05) with an IC_50_ value of 23.50 ± 3.37, 4.29 ± 0.52, and 16.99 ± 2.70 µM, respectively. This was the first report of the *α*-glucosidase inhibitory activities of compounds **3**, **9**, and **13**. Our study showed that compounds **3** and **13** have two hydroxyl groups over the ring of the pentacyclic triterpenes, which indicated the hydroxyl group in the ring of the pentacyclic triterpenes may be an effective functional group as potent *α*-glucosidase inhibitors. Previous studies also reported that the hydroxyl group of pentacyclic triterpenes has been found to confer a variety of biological properties, such as anti-tumor, anti-inflammatory, antimicrobial and hypoglycemic activities [45].

### 2.4. Enzyme Kinetic Equation

It is well accepted that enzyme kinetics can provided some useful information for predicting the interactions between the ligands and enzymes. To clarify how and where triterpenes bind to *α*-glucosidase, we first measured the enzyme kinetics of compounds **3**, **9** and **13,** by using methods similar to those described in the literature [46]. As shown in Figure 7, the concentrations of 1/[*p*NPG] are displayed on the *X*-axis, and 1/*v* values obtained from the Lineweaver–Burk plot are shown along the *Y*-axis. The plots of compound **3** intersected in the second quadrant, meaning that the Vmax values decreased and the Km values increased with the increased concentration of inhibitors (Figure 7a). The plots of compounds **9** and **13** intersected in the third quadrant, meaning that both the Km and Vmax values decreased with the increased concentration of inhibitors (Figure 7b,c). The results indicated that compounds **3**, **9** and **13** caused a mixed-type inhibition, which meant they could bind to both the free enzyme and the enzyme-substrate complex [47].

We also examined Dixon plots of how compounds **3**, **9** and **13** affect *α*-glucosidase. As shown in Figure 8, these plots further confirmed that compounds **3**, **9** and **13** are mixed-type *α*-glucosidase inhibitors. The *K_i_* values of compounds **3**, **9** and *13* were 56.86 ± 1.23, 48.88 ± 0.07 and 13.63 ± 0.42 μM, respectively, while the *K_i_*’ values of these compounds were 47.89 ± 1.37, 19.52 ± 0.26 and 8.82 ± 0.06 μM, respectively. *K_i_* is the equilibrium constant for the inhibitor binding to *α*-glucosidase, and *K_i_*’ is the equilibrium constant for the inhibitor binding to the *α*-glucosidase−*p*NPG complex [48]. The results showed that the *K_i_*’ values were smaller than the *K_i_* values, which indicated that the inhibitor−enzyme−substrate complex binding affinity exceeds the binding affinity of the inhibitor−enzyme. The binding sites and mechanism underlying inhibition have yet to be determined. However, the results of compounds **3**, **9** and **13** bound to either *α*-glucosidase, or the *α*-glucosidase−*p*NPG complex, further confirmed that compounds **3**, **9** and **13** are mixed−competitive inhibition against *α*-glucosidase.

### 2.5. Molecular Docking

Molecular docking is a key tool in structural molecular biology and computer-assisted drug design. The goal of ligand–protein docking is to predict the predominant binding modes of a ligand with a protein of known 3D structure [49]. The calculated binding energies of 2*β*,3*β*-dihydroxyolean-12-en-28-oic acid 28-*O*-*β*-D-glucopyranoside (**3**), and 22*β*-hydroxy-oleanolic acid (**13**) with *α*-glucosidase, were found to be −4.99 and −4.63 kcal/mol, respectively. But the binding energy of compound **9** with *α*-glucosidase was found to be +2.74 kcal/mol, which indicated poor binding (Figure 9). This result further suggested that compound **9** exerted the effect of inhibiting *α*-glucosidase activity by preferentially binding to the enzyme–substrate complex. This coincided with the results of the enzyme kinetic analysis, in which the *K_i_* and *K_i_*’ values of compound **9** were 48.88 ± 0.07 and 19.52 ± 0.26 μM, respectively. Interestingly, significant H-bonding interactions with the hydroxyl groups of compounds **3** and **13** were found in all these binding sites. For compound **3**, there are four residues (Asn 247, Thr 285, Ser 282 and Asp 242) which formed six hydrogen bonds with the compound. Among these, the hydrogen of the hydroxyl groups at the C-2 and C-3 position on the ring A of compound **3** formed four hydrogen-bonding interactions with Asn 247, Thr 285, Ser 282 residues of the enzyme (Figure 10). For compound **13**, only one hydrogen bond was formed between the compound and the residues of *α*-glucosidase. It was established between the hydrogen of the hydroxyl group at the C-3 position on the ring A of compound **13** and Arg 359, with a distance of 2.1 Å (Figure 10). This accounts well for the previous observation that hydroxyl groups were essential to improve the inhibitory activity of the compound.

## 3. Materials and Methods

### 3.1. Chemicals, Reagents and Cell

^1^H and ^13^C and 2D NMR spectra were obtained on a Bruker–Avance Ⅲ-500 MHz (Bruker Corporation, Madison, WI, USA) spectrometer with chemical shifts recorded in *δ* (ppm), using tetramethylsilane (TMS) as the internal standard, while the coupling constants (*J*) were given in hertz. Mass spectra were obtained on an MS Waters AutoSpec Premier P776 mass spectrometer (ESI-MS) and a UPLC-IT-TOF-MS (HR-ESI-MS), respectively. An IR spectrum was recorded on a BRUKER VERTEX 70 (Bruker Corporation, Madison, WI, USA) spectrometer in KBr pellets. The UV spectrum was measured on a SHIMADZU UV-2401PC (Shimadzu Corporation, Berlin, Germany) series spectrophotometer, with methanol as a solvent. CD spectra was obtained on an Applied Photophysics Chirascan Circular Dichroism Spectrometer (Applied Photophysics Ltd, UK). Optical rotation was taken on an Autopol VI, Serial #91058. Column chromatography was run on silica gel (80–100 mesh and 200–300 mesh) (Qingdao Marine Chemical Co., Ltd., Qingdao, China), LiChroprep RP-C18 gel (Merck, 40–63 μm, Darmstadt, Germany) and Sephadex LH-20 (Gytiva Sweden AB, Upsala, Sweden). Fractions were monitored by thin-layer chromatography (TLC), and spots were visualized by heating silica gel plates sprayed with 10% H_2_SO_4_/CH_3_CH_2_OH. A semipreparative HPLC was run on a Shimadzu system (Shimadzu Corporation, Nakagyo-ku, Kyoto, Japan) with a Shim-pack Scepter C18-120 (4.6 mm × 250 mm, 5 µm). The 3T3-L1 mouse preadipocytes were purchased from the American Type Culture Collection (ATCC, Manassas, VA, USA). High-glucose DMEM, low-glucose DMEM, Pen-Strep solution (P/S), insulin, certified fetal bovine serum (FBS), special newborn calf serum (NBCS), and phosphate buffered saline (PBS) were purchased from Biological Industries (Shanghai, China). 3-Isobutyl-1-methylxanthine (IBMX) and dexamethasone (DEX) were obtained from Sigma-Aldrich (St. Louis, MO, USA). Rosiglitazone (ROSI) was purchased from Meilun Biotech Co., Ltd. (Dalian, Liaoning, China). Dimethyl sulfoxide (DMSO) was obtained from Solarbio (Beijing, China). The glucose test kit was purchased from Rongsheng Biotech Co., Ltd. (Shanghai, China). CellTiter 96^®^ AQueous One Solution Cell Proliferation Assay was also acquired (Promega Corporation, Madison, WI, USA). *α*-Glucosidase (33 U/mg), acarbose, 4-nitrophenyl-*α*-D-glucopyranoside (*p*NPG), and ascorbic acid were purchased from Yuanye Biotech Co., Ltd. (Shanghai, China). The other chemicals and reagents were purchased from local suppliers. The absorbance was measured by a microplate reader (Molecular Devices, Palo Alto, Santa Clara, CA, USA).

### 3.2. Plant Materials

KF, the fruits of *Kochia scoparia* (Linn.) Schrad produced in Shandong, were purchased from Yunnan Lvsheng Pharmaceutical Co., Ltd (Yunnan, China). in 2019, and identified by Prof. Yumei Zhang of Xishuangbanna Tropical Botanical Garden, Chinese Academy of Sciences. A voucher specimen (No. 2019001) of KF was deposited in the Innovative Drug Research Group, Xishuangbanna Tropical Botanical Garden, Chinese Academy of Sciences.

### 3.3. Extraction and Isolation

The air-dried fruits of *K. scoparia* (20 kg) were extracted four times (7, 3, 3, 1 day, respectively) with 80% ethanol (45, 30, 30, 30 L, respectively) at room temperature. The extracts were filtered and evaporated under reduced pressure to obtain the ethanol extract of Kochiae Fructus (EE, 2.03 kg). Then, the EE (2.0 kg) was subjected to silica gel column chromatography and eluted with petroleum ether, ethyl acetate, and ethanol to obtain the petroleum ether fraction of EE (PEF, 0.15 kg), ethyl acetate fraction of EE (EAF, 0.26 kg), and ethanol fraction of EE (ETF, 1.40 kg), respectively. The ETF (1.40 kg) was subjected to silica gel column chromatography and eluted with ethyl acetate/methanol (3:1, 0:1) to yield two fractions (Fr. 1, Fr. 2). Then, the two fractions were dissolved with 20% (*v*/*v*) ethanol (EtOH)/water (H_2_O) solution and centrifuged at 5000 rpm for fifteen minutes, respectively. Two supernatants were evaporated under reduced pressure to yield two sub-fractions (Fr. 1.1, Fr. 2.1), respectively. Removal of the supernatant gave the sediment (Fr. 3) by rotary evaporators. Fr. 1.1 (114.7 g) was chromatographed on a silica gel column (200–300 mesh) and eluted with chloroform (CHCl_3_)/methanol (MeOH) gradient system (20:1 to 1:1) to yield two sub-fractions 1.1.1 and 1.1.2). Fr. 1.1.1 (74.6 g) was subjected to repeated column chromatography over silica gel, RP-C18, Sephadex LH-20, preparative HPLC, to afford compounds **3** (3.08 mg), **7** (4.21 mg), **8** (9.11 mg), and **12** (17.52 mg). Fr. 1.1.2 (42.4 g) was subjected to repeated column chromatography over silica gel, RP-C18, Sephadex LH-20 (Gytiva Sweden AB, Upsala, Sweden), preparative HPLC, to afford compounds **5** (3.04 mg), **13** (2.75 mg), **14** (16.5 mg), **15** (3.28 mg), **16** (2.05 mg), and **17** (5.37 mg). Fr. 2.1 (100 g) was separated by Sephadex LH-20 column using CHCl_3_-MeOH isocratic system (1:1) to afford three sub-fractions (Fr. 2.1.1, Fr. 2.1.2, and Fr. 2.1.3). Fr. 2.1.1 (27.7 g) was subjected to repeated column chromatography over RP-C18 and a preparative HPLC to afford compounds **1** (5.03 mg), **2** (2.86 mg), **9** (3.28 mg) and **11** (2.46 mg). Fr. 3 (100 g) was ground and dissolved by heating with 100% ethanol solution at 60 °C, filtered, and the ethanol in the filtrate was removed under reduced pressure to obtain sub-fraction 3.1 (Fr. 3.1). Fr. 3.1 (86 g) was ground into powder and dissolved with ethanol (86 × 5 mL); the mixture solution was stirred using a magnetic stirrer at 600 rpm for 24 h. Finally, the filter residue was obtained by filtration. After repeating the above operation four times, the filter residue was finally obtained (Fr. 3.1.1, 30 g). Fr. 3.1.1 (30 g) was purified by a preparative HPLC (MeOH-H_2_O-acetic acid (CH_3_COOH), 85:15:0.2, isocratic) to give compound **4** (5.2 g). The EAF (260 g) was subjected to silica gel column chromatography and eluted with CHCl_3_/MeOH (50:1 to 0:1) to yield compound **6** (3.37 mg). The PEF (150 g) was subjected to silica gel column chromatography and eluted with petroleum ether/CHCl_3_ (50:1 to 0:1) to yield compound **10** (56.1 g).

#### 3.3.1. 3(β)-3-O-β-D-xylopyranosly(1→3)-β-D-glucopyranosiduronic acid-12-en-23,28-dioic acid (**1**)

The molecular formula was C_41_H_62_O_15_, white amorphous powder; [α]25 D + 17.1° (c 0.187, MeOH); UV/Vis (MeOH, λ_max_, nm) (log ε): 203 (3.92). IR (KBr) *v*_max_ 3434, 2928, 2870, 1625, 1385, 1038 cm^−1^; ECD (MeOH, λ_max_, nm) (△ε): 195 (8.22), 262 (0.28). ^1^H NMR (500 MHz, methanol-*d*_4_) and ^13^C NMR (125 MHz, methanol-*d*_4_), see Table 1 and Table 2; HR-ESI-MS *m*/*z* 396.1971 [M-2H]^2-^ (calcd 396.1972) (see Appendix A).

#### 3.3.2. 3(β)-3-*O*-*β*-D-xylopyranosly(1→3)-*β*-D-glucopyranosiduronic acid 6-methyl ester-12-en-23,28-dioic acid (**2**)

The molecular formula was C_42_H_64_O_15_, white amorphous powder; [α]25 D + 4.5° (c 0.089, MeOH); UV/Vis (MeOH, λ_max_, nm) (log ε): 203 (3.92). IR (KBr) *v*_max_ 3419, 2927, 2864, 1694, 1386, 1258, 1037 cm^−1^; ECD (MeOH, λ_max_, nm) (△ε): 195 (4.62), 262 (0.23). ^1^H NMR (500 MHz, methanol-*d*_4_) and ^13^C NMR (125 MHz, methanol-*d*_4_), see Table 1 and Table 2; HR-ESI-MS *m*/*z* 403.2053 [M-2H]^2-^ (calcd 403.2050) (see Appendix A).

#### 3.3.3. 2β,3β-dihydroxyolean-12-en-28-oic acid 28-*O*-*β*-D-glucopyranoside (**3**)

The molecular formula was C_36_H_58_O_9_, white amorphous powder; [α]25 D + 22.67° (c 0.172, MeOH); UV/Vis (CH_3_OH, λ_max_, nm) (log ε): 203 (3.76). IR (KBr) *v*_max_ 3422, 2928, 2867, 1738, 1385, 1259, 1071, 1030 cm^−1^; ECD (MeOH, λ_max_, nm) (△ε): 195 (6.31), 241 (0.40). ^1^H NMR (500 MHz, methanol-*d*_4_) and ^13^C NMR (125 MHz, methanol-*d*_4_), see Table 1 and Table 2; HR-ESI-MS *m*/*z* 657.3973 [M+Na]^+^ (calcd 657.3973) (see Appendix A).

### 3.4. 3T3-L1 Preadipocytes Culture and Differentiation

The murine 3T3-L1 preadipocytes were supplied by the American Type Culture Collection (ATCC, Manassas, VA, USA). The cells were cultured in high-glucose DMEM supplemented with 10% (*v*/*v*) newborn calf serum (NBCS), 1% (*v*/*v*) Pen-Strep solution (P/S) at 37 °C in a humidified atmosphere of 5% CO_2_, and then starved until the cells reached confluence (day 0). Two days later (day 2), the cells were cultured in DMEM supplemented with 10% (*v*/*v*) fetal bovine serum (FBS), 1% (*v*/*v*) P/S, 0.5 mM 3-isobutyl-1-methylxanthine (IBMX), 1 μM dexamethasone (DEX), 1 μM rosiglitazone (ROSI) and 100 nM insulin. Three days after (day 5) the induction, the culture medium was replaced with DMEM containing 10% (*v*/*v*) FBS, 1% (*v*/*v*) P/S, and 100 nM insulin for one day (day 6). Then the cells were completely differentiated into mature adipocytes (Figure 11).

### 3.5. Glucose Uptake and Cell Viability Assays

The mature adipocytes were inoculated in a 96-well plate at 5 × 104 cells per well for 24 h. The 3T3-L1 adipocytes were divided into model groups (blank control), insulin group (250 ng/mL, positive control), and sample group (20 μM). After 24 h of administration, 10 μL medium was used to measure the glucose content. The absorbance value was detected by a microplate reader at 505 nm. The glucose uptake rate was calculated as follows:Glucose uptake rate (%) = [1 − A_2_/A_1_] × 100(1)
where A_1_ is the absorbance of the mixed-glucose, DMEM-supplemented control, and A_2_ is the absorbance of the blank, insulin, or sample group.

Cell viability was detected by using a CellTiter 96^®^ AQueous One Solution Cell Proliferation Assay (Promega Corporation, Madison, WI, USA) according to the manufacturer’s instructions after the glucose uptake experiment [50]. CellTiter 96^®^ AQueous One Solution Cell Proliferation Assay reagent (20 µL/well) was added to the plate and incubated at 37 °C for 180 min before absorbance was measured at 490 nm, and the relative cell viability was calculated as follows:Relative cell viability (%) = A_1_/A_2_ × 100(2)
where A_1_ is the absorbance of the blank control, and A_2_ is the absorbance of the sample group.

### 3.6. In Vitro α-Glucosidase Inhibition Assay

The *α*-glucosidase inhibition assay was carried out on the basis of the method reported by Zhang et al., with minor modifications [51]. Briefly, *α*-glucosidase (0.1 U/mL), *p*NPG (5 mM) and Na_2_CO_3_ (0.5 M) were prepared in PBS (0.1 M, pH 6.8), and the samples were diluted to different concentrations (0.0625, 0.125, 0.25, 0.5, 1, and 2 mM) using PBS. In a 96-well microplate, a mixture of 80 μL PBS, 10 μL sample, and 50 μL *α*-glucosidase solution were added and incubated for 15 min at 37 °C, 10 μL PBS used as a blank control. To initiate the reaction, *p*NPG (40 μL) was added to the reaction mixture and incubated at 37 °C for 30 min. The reaction was terminated by adding 20 μL of Na_2_CO_3_, after which the absorbance was determined at 405 nm by a microplate reader (SpectraMax190, Micro-g Biotech, Guangzhou, China). Acarbose was used as a positive control in this *α*-glucosidase inhibition assay. IC_50_ values were defined as the concentration of the compound required to inhibit 50% of *α*-glucosidase activity under assay conditions. The *α*-glucosidase inhibition activity was calculated as follows:Inhibition rate (%) = (A_1_ − A_2_)/A_1_ × 100(3)
where A_1_ is the OD value of the blank control, A_2_ is the OD value of the tested samples, and the analysis was performed in triplicates.

### 3.7. Kinetics Involved in the Inhibition of α-Glucosidase

The kinetic analysis of compounds **3**, **9**, and **13** were measured using the reaction conditions in Section 3.6. Typically, three different concentrations of each compound around the IC_50_ values were chosen. Under each concentration, *α*-glucosidase activity was assayed by varying the concentration of *p*NPG as a substrate [46]. The inhibition types of active compounds were determined by Lineweaver–Burk plots [the inverse of velocity (1/*v*) against the inverse of the substrate concentration (1/[*p*NPG])] with substrate concentrations of 1.25, 2.5, 5, 10, 20 µM. *K_i_* and *K_i_*’ values were determined from 1/*v* versus [I] (Dixon plot) and S/*v* versus [I] plots, respectively.

### 3.8. Molecular Docking

The molecular docking approach can be used to model the interaction between a small molecule and a protein at the atomic level [52]. The structure of *α*-glucosidase (PDB ID: 3A4A) was obtained from the Online Protein Data Bank [53], and the 3D structures of the ligands were generated by Chem3D Pro (version: 14.0). Complexed ligands and water molecules in the crystal structure of *α*-glucosidase were virtually removed by PyMOL Win application (PyMOL, version: 2.4.0). Gasteiger charges and essential hydrogen atoms were added by using the AutoDock tools (ADT, version: 1.5.6). The cubic grid box dimensions of *α*-glucosidase were defined as x = 98, y = 126, and z = 102 Å with spacing of 0.692 Å. Finally, the PyMOL molecular graphics system (version 2.4.0) was used to visualize ligand–enzyme interactions.

### 3.9. Statistical Analysis

IBM SPSS Statistics for Windows, version 26.0 (IBM Corp., Armonk, NY, USA), was used to analyze all of the data. The experiments were carried out in triplicates and the results were expressed as an average of the three measurements ± SD. One-way analysis of variance (ANOVA) was used to compare the means of different analysis investigations. Differences were considered significant when * *p* < 0.05, ** *p* < 0.01, *** *p* < 0.001.

## 4. Conclusions

Seventeen compounds, including three previously undescribed and fourteen known triterpenes, were isolated from the ethanol extract of KF. We detected their hypoglycemic activities via assays for *α*-glucosidase inhibition and glucose uptake of 3T3-L1 adipocytes. Next, we performed enzyme kinetics and molecular docking investigations to analyze the possible mechanisms against enzymes. The results of the glucose uptake experiment showed compound **13** had a significant promotion on glucose uptake rate of 3T3-L1 adipocytes (*p* < 0.001). Simultaneously, enzyme-inhibition results suggested that compounds **3**, **9**, and **13** possessed potent inhibitory effects on *α*-glucosidase, and their enzymatic kinetics on *α*-glucosidase showed that they are mixed-type inhibitors. The hydroxyl group in the ring of the pentacyclic triterpene played a key role in maintaining *α*-glucosidase inhibitory activity according to the docking simulation. In summary, this study enriched the chemical composition diversity of KF and provided effective evidence for its use in hypoglycemic herbal medicine.

## Figures and Tables

**Figure 1 ijms-24-02454-f001:**
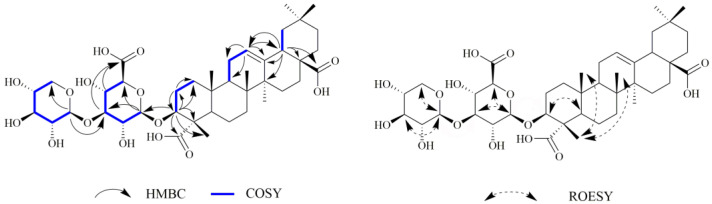
Key HMBC, COSY, and ROESY correlations of compound **1**.

**Figure 2 ijms-24-02454-f002:**
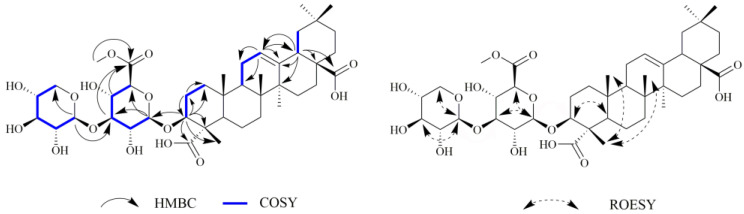
Key HMBC, COSY, and ROESY correlations of compound **2**.

**Figure 3 ijms-24-02454-f003:**
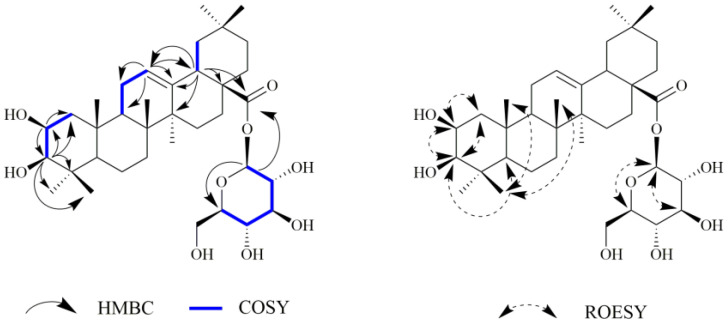
Key HMBC, COSY, and ROESY correlations of compound **3**.

**Figure 4 ijms-24-02454-f004:**
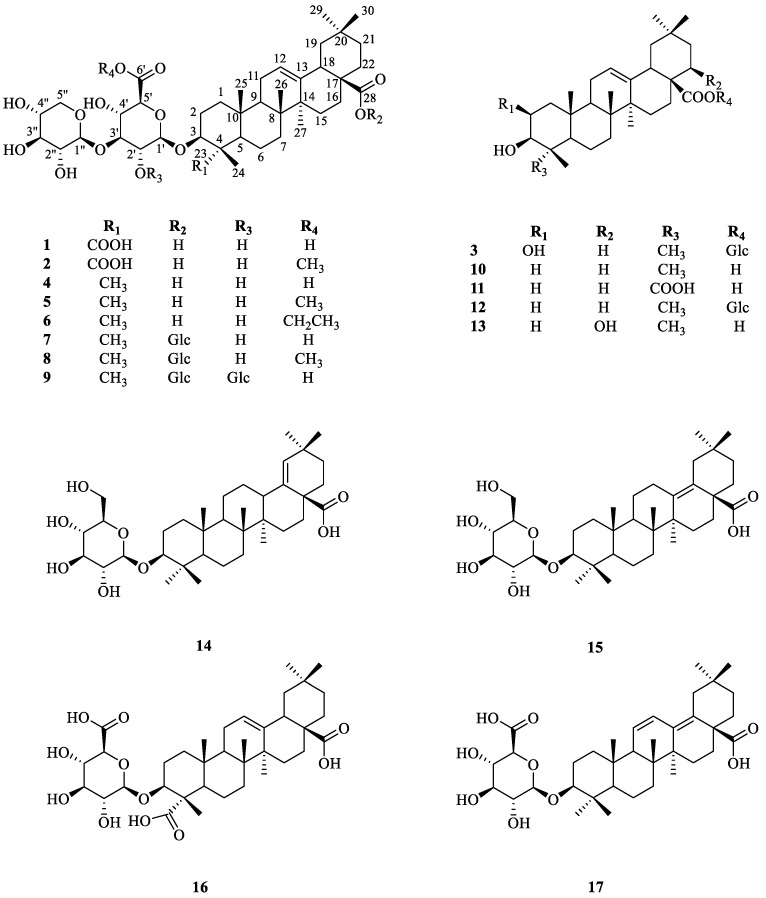
Chemical structures of compounds **1**−**17** isolated from Kochiae Fructus.

**Figure 5 ijms-24-02454-f005:**
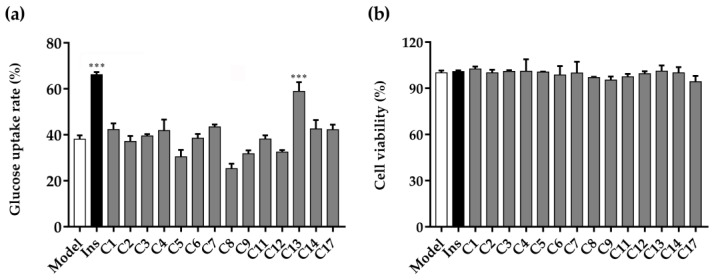
Glucose uptake and cell viability in 3T3-L1 adipocytes. (**a**) Glucose uptake rate of compounds **1**−**9**, **11**−**14**, and **17**. (**b**) Cell viability of compounds **1**−**9**, **11**−**14**, and **17**. Ins, insulin (positive control); C, compound; compounds at 20 μM and insulin at 250 ng/mL; all values are mean ± SD from a least three independent experiments, and each group is compared with model group, significance is denoted by symbols: *** *p* < 0.001.

**Figure 6 ijms-24-02454-f006:**
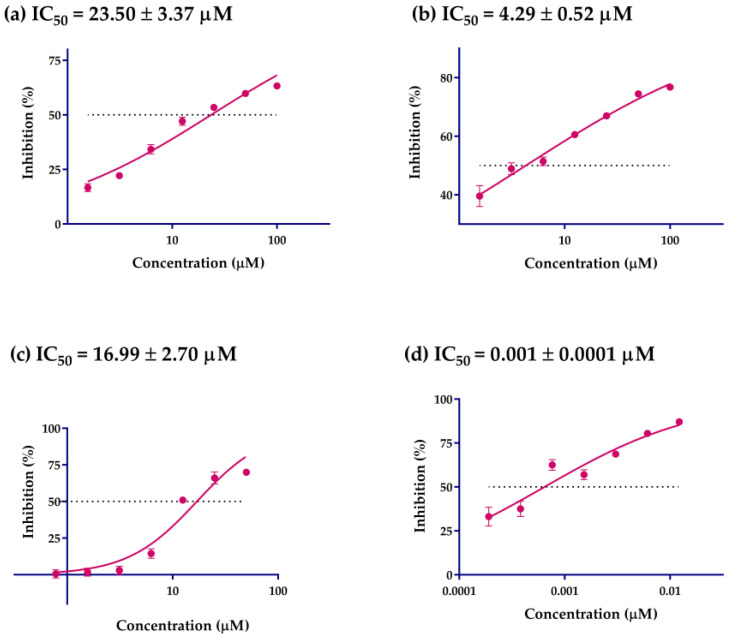
*α*-Glucosidase inhibitory effects of compounds **3**, **9**, and **13**. (**a**) Log concentration–inhibition rate fitting curve of compound **3**. (**b**) Log concentration–inhibition rate fitting curve of compound **9**. (**c**) Log concentration–inhibition rate fitting curve of compound **13**. (**d**) Log concentration–inhibition rate fitting curve of acarbose. Calculated IC_50_ values of different groups by Statistical Product and Service Solutions (SPSS, version: 21.0), and all values are mean ± SD from a least three independent experiments.

**Figure 7 ijms-24-02454-f007:**
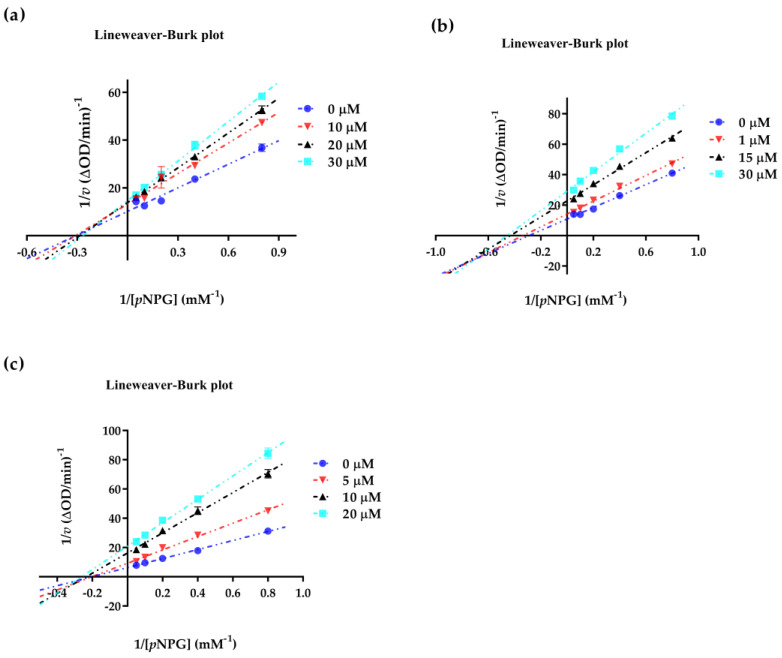
The Lineweaver–Burk plot for *α*-glucosidase inhibition of compounds **3**, **9**, and **13**. (**a**) 1/[*p*NPG]−1/*v* fitting curve of compound **3**. (**b**) 1/[*p*NPG]−1/*v* fitting curve of compound **9**. (**c**) 1/[*p*NPG]−1/*v* fitting curve of compound **13**. All values are mean ± SD from at least three independent experiments.

**Figure 8 ijms-24-02454-f008:**
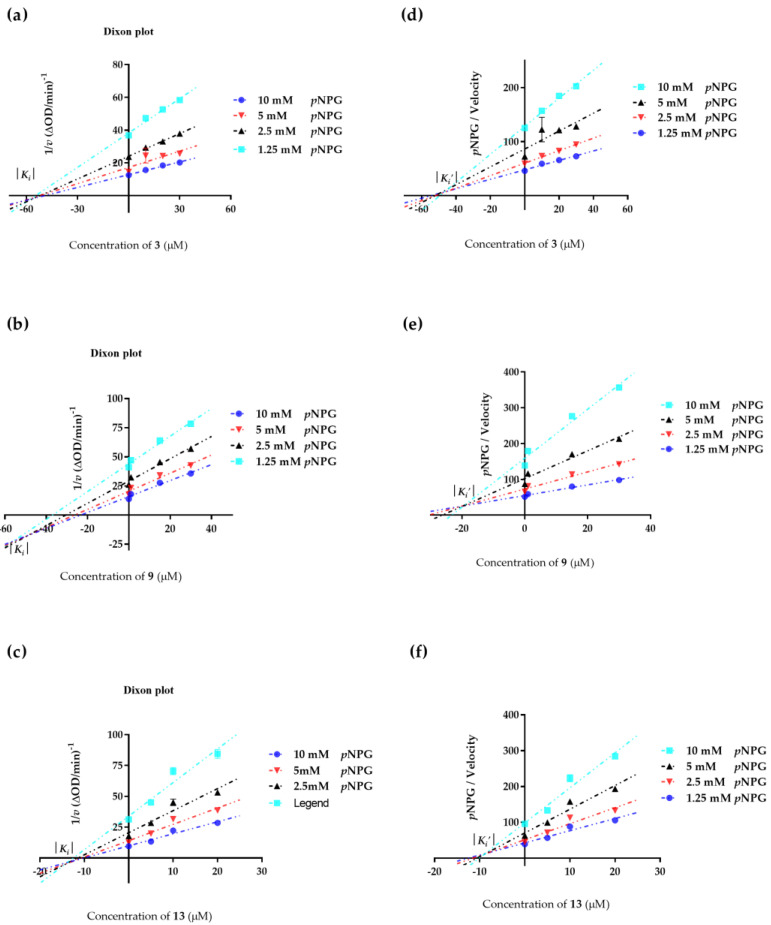
Determination of *K_i_* and *K_i_*’ of compounds **3**, **9**, and **13** on *α*-glucosidase. (**a**) 1/*v*−[I] fitting curve of compound **3**. (**b**) 1/*v*−[I] fitting curve of compound **9**. (**c**) 1/*v*−[I] fitting curve of compound **13**. (**d**) [*p*NPG]/*v*−[I] fitting curve of compound **3**. (**e**) [*p*NPG]/*v*−[I] fitting curve of compound **9**. (**f**) [*p*NPG]/*v*−[I] fitting curve of compound **13**. All values are mean ± SD from at least three independent experiments.

**Figure 9 ijms-24-02454-f009:**
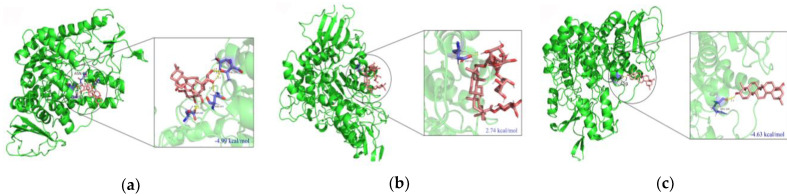
Molecular docking pictures of compounds **3**, **9**, and **13** on *α*-glucosidase of 3A4A. (**a**) 3D-structural model of α-glucosidase bound to compound **3** and the close-up view of the compound **3** molecule bound at the active site of *α*-glucosidase. (**b**) 3D-structural model of *α*-glucosidase bound to compound **9** and the close-up view of the compound **9** molecule bound at the active site of *α*-glucosidase. (**c**) 3D-structural model of *α*-glucosidase bound to compound **13** and the close-up view of the compound **13** molecule bound at the active site of *α*-glucosidase. The 3D-structural model of *α*-glucosidase is shown in green; residues that may be involved in the interactions of compound-binding are drawn with a stick model and shown in purple. The possible hydrogen-bond interactions are indicated with dashed lines (yellow).

**Figure 10 ijms-24-02454-f010:**
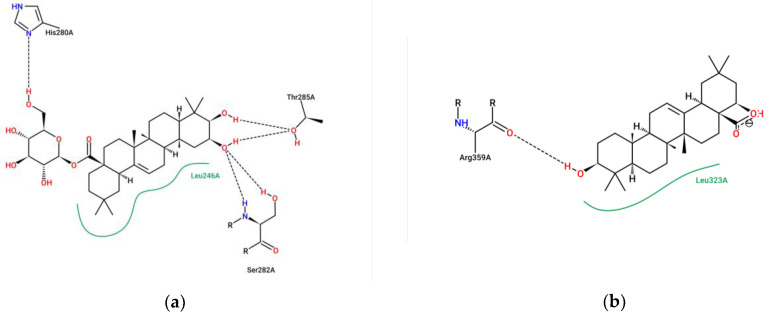
The 2D interaction diagram of the docking poses of compounds **3** (**a**) and **13** (**b**) on *α*-glucosidase of 3A4A.

**Figure 11 ijms-24-02454-f011:**
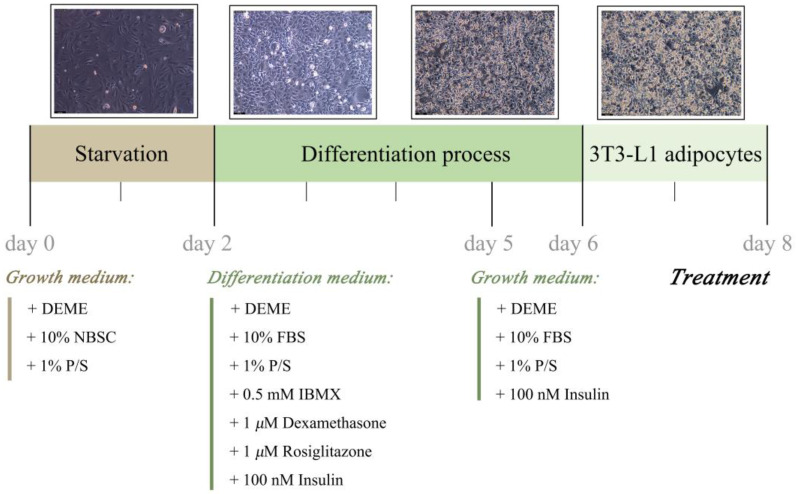
Differentiation process schedule. DMEM, dulbecco’s modified eagle medium; NBCS, special newborn calf serum; P/S, penicillin/streptomycin; FBS, certified fetal bovine serum; IBMX, 3-isobutyl-1-methyl-xanthine.

**Table 1 ijms-24-02454-t001:** ^1^H NMR (500 MHz) and ^13^C NMR (125 MHz) data of compounds **1**–**3** in methanol-*d*_4_.

No.	1		2		3	
	*δ*_H_, (*J* in Hz)	*δ* _C_	*δ*_H_, (*J* in Hz)	*δ* _C_	*δ*_H_, (*J* in Hz)	*δ* _C_
1	1.67, overlapped	39.7	1.66, overlapped	39.7	1.92, m	48.3
	1.08, overlapped		1.08, overlapped		0.88, m	
2	1.87, m	26.6	1.83, m	26.6	3.62, m	69.6
	1.68, m		1.67, m			
3	4.05, dd (11.5, 4.0)	86.6	4.05, dd (11.5, 4.5)	86.6	2.90, d (9.5)	84.6
4		54.0		54.1		40.7
5	1.49, m	53.1	1.49, m	53.1	0.84, m	56.9
6	2.00, td (13.5, 3.5)	24.2	2.01, td (13.5, 4.0)	24.2	2.04, td (13.5, 3.5)	24.1
	1.59, m		1.60, m		1.70, m	
7	1.53, overlapped	33.7	1.53, overlapped	33.7	1.56, overlapped	34.0
	1.53, overlapped		1.53, overlapped		1.56, overlapped	
8		41.0		41.0		40.9
9	1.65, overlapped	49.1	1.64, overlapped	49.1	1.62, overlapped	49.2
10		37.6		37.6		39.4
11	1.92, m	24.6	1.93, m	24.6	1.92, m	24.8
	1.90, m		1.91, m		1.90, m	
12	5.23, brs	123.6	5.24, brs	123.6	5.27, brs	123.8
13		145.3		145.6		145.2
14		43.0		43.1		43.1
15	1.76, overlapped	29.0	1.76, overlapped	29.0	1.79, overlapped	29.0
	1.05, overlapped		1.05, overlapped		1.05, overlapped	
16	1.57, overlapped	21.9	1.57, overlapped	21.9	1.52, overlapped	19.7
	1.57, overlapped		1.57, overlapped		1.52, overlapped	
17		47.8		47.8		48.2
18	2.84, dd (13.5, 4.0)	42.9	2.85, dd (14.0, 4.0)	42.9	2.86, dd (14.0, 4.0)	42.7
19	1.71, m	47.4	1.69, m	47.4	1.70, m	47.4
	1.11, m		1.11, m		1.13, m	
20		31.8		31.8		31.7
21	1.38, td (13.5, 3.5)	35.0	1.39, td (13.5, 4.0)	35.0	1.39, m	35.0
	1.21, m		1.22, m		1.21, m	
22	1.73, overlapped	33.9	1.72, overlapped	34.0	1.72, overlapped	33.3
	1.27, overlapped		1.27, overlapped		1.32, overlapped	
23		181.5		181.7	1.01, s	29.4
24	1.14, s	12.2	1.14, s	12.4	0.80, s	17.9
25	0.96, s	16.4	0.96, s	16.4	1.01, s	17.3
26	0.80, s	17.7	0.81, s	17.7	0.80, s	17.6
27	1.16, s	26.4	1.17, s	26.5	1.16, s	26.5
28		181.9		182.0		178.2
29	0.90, s	33.7	0.91, s	33.7	0.91, s	33.6
30	0.93, s	24.1	0.94, s	24.1	0.94, s	24.1

**Table 2 ijms-24-02454-t002:** ^1^H NMR (500 MHz) and ^13^C NMR (125 MHz) data of the sugar moieties for compounds **1**–**3** in methanol-*d*_4_.

No.	1		2		3	
	*δ*_H_, (*J* in Hz)	*δ* _C_	*δ*_H_, (*J* in Hz)	*δ* _C_	*δ*_H_, (*J* in Hz)	*δ* _C_
GlcA-at-C-3					Glc-at-C-28	
1′	4.37, d (7.5)	105.5	4.39, d (7.5)	105.5	5.38, d (8.0)	95.9
2′	3.35, d (7.5)	74.6	3.34, d (7.5)	74.6	3.31, m	74.1
3′	3.50, m	86.3	3.52, m	86.1	3.40, m	78.5
4′	3.56, m	71.8	3.57, m	71.7	3.34, m	71.3
5′	3.76, m	76.4	3.83, d (9.5)	76.5	3.34, m	78.9
6′		171.4		171.2	3.81, dd (12.0, 1.5)	62.6
					3.68, dt (12.0, 2.0)	
6′-OMe			3.77, s	53.0		
Xyl→glc-C-3						
1″	4.50, d (7.5)	105.9	4.52, d (7.5)	105.9		
2″	3.24, m	75.4	3.24, m	75.4		
3″	3.32, m	77.7	3.32, m	77.7		
4″	3.47, m	71.2	3.49, m	71.2		
5″	3.89, dd (11.5, 5.5)	67.2	3.89, dd (11.5, 5.5)	67.2		
	3.20, dd (11.5, 5.5)		3.20, dd (11.5, 5.5)			

**Table 3 ijms-24-02454-t003:** *α*-Glucosidase inhibitory activities of compounds **1**−**9**, **11**−**14**, and **17**.

Compounds	Inhibitory Rate (%)	IC_50_ (μM)
Acarbose ^A^	87.04 ± 0.29 ^a^	0.001 ± 0.00
**1** **^B^**	13.71 ± 0.54 ^j^	>50
**2** **^B^**	14.05 ± 4.90 ^j^	>50
**3** **^B^**	59.79 ± 1.04 ^d^	23.50 ± 3.37
**4** **^B^**	26.28 ± 1.01 ^i^	>50
**5** **^B^**	29.62 ± 2.26 ^i^	>50
**6** **^B^**	35.00 ± 0.40 ^h^	>50
**7** **^B^**	52.71 ± 0.85 ^f^	>50
**8** **^B^**	57.02 ± 36.3 ^e^	>50
**9** **^B^**	74.41 ± 1.02 ^b^	4.29 ± 0.52
**11** **^B^**	53.98 ± 1.65 ^e^	>50
**12** **^B^**	41.48 ± 1.66 ^g^	>50
**13** **^B^**	69.91 ± 0.59 ^c^	16.99 ± 2.70
**14 ^B^**	17.49 ± 4.45 ^j^	>50
**1** **7 ^B^**	55.68 ± 0.17 ^e^	>50

Data were expressed as the mean value ± SD (*n* = 3); means followed by the different letters (^a−j^) are significantly different (*p* < 0.05); ^A^, percent inhibition at a concentration of 0.012 μM; ^B^, percent inhibition at a concentration of 50 μM.

## Data Availability

All data presented in this study are available in the article.

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
