# Peer review of "Triterpenoids from Kochiae Fructus: Glucose Uptake in 3T3-L1 Adipocytes and α-Glucosidase Inhibition, In Silico Molecular Docking"

_ijms, 2023, doi:10.3390/ijms24032454_

Round 1

Reviewer 1 Report

Dear author,

The article: Triterpenoids from Kochiae Fructus: glucose uptake in 3T3-L1 adipocytes and α-glucosidase inhibition, in silico molecular docking is appropriate to publish in this Journal if the author make some following major revision

1. It would be great if the author gave more information about the relationship between alpha-glucosidase, glucose uptake, and docking experiment results. How do they relate the results of their experiment?

2. In Table 3, why is the inhibition rate of acarbose not determined? The inhibition rate of positive control should be provided.

3. Please describe how the plant fruits were stored and in what form they were analyzed because the samples have been collected since 2019.

4. Please describe the abbreviation that the author used in the extraction and isolation section before using it.

5. Did the author try to crystallize all of the pure compounds? because the structure can form crystals easily in MeOH or DCM under refrigeration.

6. The ECD data of all compounds should be identified in the supporting information to confirm their absolute configuration.

7. Please give more information why the author used different days and liter of solvent at the extraction process? 

8. Did the author check and confirm the purity of pure compounds before biological testing? Please check the purity of the test compounds.

Author Response

Dear professor,

Thank you for your kind help to review our manuscript. we really appreciate all of your positive and constructive comments and suggestions on our manuscript entitled “Triterpenoids from Kochiae Fructus: glucose uptake in 3T3-L1 adipocytes and α-glucosidase inhibition, in silico molecular docking”. According to your valuable suggestions, we have revised the manuscript carefully and responded as follows: 

1.It would be great if the author gave more information about the relationship between α-glucosidase, glucose uptake, and docking experiment results. How do they relate the results of their experiment?
Response: The 3T3-L1 adipocyte glucose uptake model and in vitro α-glucosidase inhibition assay are effective and feasible methods for type 2 diabetes mellitus research, and their pathways for hypoglycemic effects are different. The glucose uptake model of 3T3-L1 adipocytes exerts hypoglycemic effects by enhancing the glucose uptake of adipocytes, however, in vitro α-glucosidase inhibition assay exerts hypoglycemic effects by retarding the absorption of glucose in the small intestine and decreasing the postprandial blood glucose levels. The results of molecular docking are to predict the predominant binding modes between ligands with α-glucosidase. 

2.In Table 3, why is the inhibition rate of acarbose not determined? The inhibition rate of positive control should be provided.
Response: In this experiment, the inhibition rate of acarbose was determined. When the concentration of acarbose was 0.012 μM, the inhibition rate of acarbose was 87%. We have supplemented the activity data of acarbose in Table 3.

3.Please describe how the plant fruits were stored and in what form they were analyzed because the samples have been collected since 2019.
Response: The air-dried fruits of K. scoparia were extracted with 80% ethanol to obtain the ethanol extract of KF in 2019, and the chemical separation of it was completed in 2020. The literature survey and data collation were completed recently. The separated compounds are stored in brown medicine bottles and stored in a refrigerator at 4℃. And 100 g of the plant fruits was left as a voucher specimen and stored in a brown sealed bottle at room temperature.

4.Please describe the abbreviation that the author used in the extraction and isolation section before using it.
Response: We supplemented the abbreviation in the extraction and isolation section, as follow: “the ethanol extract of Kochiae Fructus (EE); the petroleum ether fraction of EE (PEF); the ethyl acetate fraction of EE (EAF); the ethanol fraction of EE (ETF); ethanol (EtOH) / water (H2O); chloroform (CHCl3) / methanol (MeOH); acetic acid (CH3COOH)”.

5.Did the author try to crystallize all of the pure compounds? because the structure can form crystals easily in MeOH or DCM under refrigeration.
Response: Thank you very much for your valuable suggestions, crystallization is indeed a great way to purify compounds and identify their structures by X-ray. In this work, we mainly purified compounds by separation methods such as silica gel, RP-C18, Sephadex LH-20, preparative HPLC. In future work, we will try to crystallize the compounds.

6.The CD data of all compounds should be identified in the supporting information to confirm their absolute configuration.
Response: For three new compounds (1-3), we supplemented their CD data to confirm their absolute configuration. The CD data of three new compounds have been added to section 3.3.1, section 3.3.2, and section 3.3.3. As follows: “Compound 1: CD (MeOH, λmax, nm) (â–³ε): 195 (8.22), 262 (0.28). Compound 2: CD (MeOH, λmax, nm) (â–³ε): 195 (4.62), 262 (0.23). Compound 3: CD (MeOH, λmax, nm) (â–³ε): 195 (6.31), 241 (0.40).”. The CD spectra figures of compounds have been added to “Supplementary material”.  As follows:

Figure S33. The CD spectra of compounds 1, 2, and 3 in MeOH.
For known compounds, their configurations were identified mainly by data from published literatures and NMR data. 

7.Please give more information why the author used different days and liter of solvent at the extraction process?
Response: The reasons for using different days in the extraction process: When the sample is first extracted, the extraction time is longer, because the fruit needs to be fully soaked with ethanol, thereby destroying plant tissues such as peels, so that the chemical components in the plant are fully extracted. With the increase of the number of extractions, a large number of chemical components in the plant have been extracted, so the soaking time is shortened when extracted later, which further improves the work efficiency.
The reasons for using different liter of solvent in the extraction process: Extract the sample in the same glass container, the first time used more solvent, and the later times used less solvent, because the sample is dry in the first time and can hold more solvent, and the sample remained residual solvent left in the later times, so it did'n need that much solvent as the first time.

8.Did the author check and confirm the purity of pure compounds before biological testing? Please check the purity of the test compounds.
Response: Yes, we did,the purity of the test compounds is all above 95%. 

Reviewer 2 Report

This paper deals with the hypoglycemic potential testing of triterpenes isolated from the ethanol extract of Kochiae Fructus.  The work includes isolation of compounds, confirmation of their structure, hypoglycemic testing activities via assays for α-glucosidase inhibition and glucose uptake of 3T3-L1 adipocytes. Finally, enzyme kinetics and molecular docking investigations were performed to analyze possible mechanisms against enzymes. The experiments are set up properly and the work is clearly written. My only suggestion is to increase the quality of Fig 6 and Fig 7.

Author Response

Dear professor,

Thank you for your kind help to review our manuscript. we really appreciate all of your positive and constructive comments and suggestions on our manuscript entitled “Triterpenoids from Kochiae Fructus: glucose uptake in 3T3-L1 adipocytes and α-glucosidase inhibition, in silico molecular docking”. According to your valuable suggestions, we have revised the manuscript carefully and responded as follows:

  1. This paper deals with the hypoglycemic potential testing of triterpenes isolated from the ethanol extract of Kochiae Fructus. The work includes isolation of compounds, confirmation of their structure, hypoglycemic testing activities via assays for α-glucosidase inhibition and glucose uptake of 3T3-L1 adipocytes. Finally, enzyme kinetics and molecular docking investigations were performed to analyze possible mechanisms against enzymes. The experiments are set up properly and the work is clearly written. My only suggestion is to increase the quality of Fig 6 and Fig 7.

Response: By carefully checking all the figures in the paper, we believe that the quality of Fig 7 and Fig 8 should be improved. So we have improved the resolutions of Fig 7 and Fig 8, increasing the resolutions of figures from 300 dpi to 1200 dpi.

Reviewer 3 Report

Comments:

1. The sugar moieties of all new compounds (1‒3) should be acid hydrolysis to clearly identify the absolute configuration.

2. The reference of protein PDB ID: 3A4A is required. The authors should cite this reference in the revised manuscript.

3. Lines 167-186: the Physical Properties of new compounds should move to section 3.3.

4. Figure 9: The 2D interaction diagram of the docking poses of compounds 3, 9, and 13 should be added to the revised manuscript to make it clearer for the reader.

Author Response

Dear professor,

Thank you for your kind help to review our manuscript. we really appreciate all of your positive and constructive comments and suggestions on our manuscript entitled “Triterpenoids from Kochiae Fructus: glucose uptake in 3T3-L1 adipocytes and α-glucosidase inhibition, in silico molecular docking”. According to your valuable suggestions, we have revised the manuscript carefully and responded as follows:

  1. The sugar moieties of all new compounds (1‒3) should be acid hydrolysis to clearly identify the absolute configuration.

Response: Thank you very much for your suggestion, we also think that acid hydrolysis is a good way to prove the absolute configurations of the sugar moieties of all new compounds (1‒3). However, due to the low quantities of the separated compounds (1‒3) and the consumption of samples in the activity assay screening, we are very sorry that the the remaining compounds are not sufficient for acid hydrolysis.

  1. The reference of protein PDB ID: 3A4A is required. The authors should cite this reference in the revised manuscript.

Response: Thank you very much for your valuable suggestions, we have cited the reference of protein PDB ID: 3A4A in the revised manuscript. As follow: “53. Yamamoto, K.; Miyake, H.; Kusunoki, M.; Osaki, S. Crystal structures of isomaltase from Saccharomyces cerevisiae and in complex with its competitive inhibitor maltose. The FEBS Journal. 2010, 277, 4205–4214. doi: https://doi.org/10.1111/j.1742-4658.2010.07810.x.”.

  1. Lines 167-186: the Physical Properties of new compounds should move to section 3.3.

Response: The physical properties of new compounds have been moved to section 3.3, thank you.

  1. Figure 9: The 2D interaction diagram of the docking poses of compounds 3, 9, and 13 should be added to the revised manuscript to make it clearer for the reader.

Response: The 2D interaction diagram of the docking poses of compounds 3 and 13 have been added to the revised manuscript (Figure 10). The binding energy of compound 9 with α-glucosidase was found to be +2.74 kcal/mol, which indicated that compound 9 had no potential binding capacity to α-glucosidase. And compound 9 exerted the effect of inhibiting α-glucosidase activity by preferentially binding to enzyme–substrate complex. Therefore, the 2D interaction diagram of the docking poses of compounds 9 is not provided.

Round 2

Reviewer 1 Report

Dear author 

One more thing that you can improve your results of alpha-glucosidase inhibition is adding the IC50 values of active compounds that have inhibition rate more than 70%.

Author Response

Thank you very much for your valuable suggestions, we have improved our results of alpha-glucosidase inhibition. Thank you!

As follow:

Table 3. α-Glucosidase inhibitory activities of compounds 19, 1114, and 17.

Compounds

Inhibitory Rate (%)

IC50 (μM)

Acarbose A

87.04 ± 0.29 a

0.001 ± 0.00

1 B

13.71 ± 0.54 j

> 50

2 B

14.05 ± 4.90 j

> 50

3 B

59.79 ± 1.04 d

23.50 ± 3.37

4 B

26.28 ± 1.01 i

> 50

5 B

29.62 ± 2.26 i

> 50

6 B

35.00 ± 0.40 h

> 50

7 B

52.71 ± 0.85 f

> 50

8 B

57.02 ± 36.3 e

> 50

9 B

74.41 ± 1.02 b

4.29 ± 0.52 

11 B

53.98 ± 1.65 e

> 50

12 B

41.48 ± 1.66 g

> 50

13 B

69.91 ± 0.59 c

16.99 ± 2.70 

14 B

17.49 ± 4.45 j

> 50

17 B

55.68 ± 0.17 e

> 50

Data were expressed as the mean value ± SD (n = 3); Means followed by the different letters (a–j) are significantly different (p < 0.05); A, percent inhibition at a concentration of 0.012 μM; B, percent inhibition at a concentration of 50 μM.
